

**Autoregressive Statistical Modeling of a Peru Margin Multi-Proxy Holocene Record Shows**
**Correlation Not Causation, Flickering Regimes and Persistence**
**Seonmin Ahn[1], Baylor Fox-Kemper[2], Timothy Herbert[2], Charles Lawrence[1]**
[1]Division of Applied Mathematics, Brown University
[2]Dept. of Earth, Environmental, and Planetary Sciences, Brown University
*Correspondence Charles Lawrence
charles_lawrence@brown.edu
Division of Applied Mathematics.
Brown University
182 George Street
Providence, RI 02912
**Abstract**
Correlation does not necessarily imply a causation, but in climatology and paleoclimatology, correlation
is used to identify potential cause-and-effect relationships because linking mechanisms are difficult to
observe. Confounding by an often unknown outside variable that drives the sets of observables is one of
the major factors that lead to correlations that are not the result of causation. Here we show how
autoregressive (AR) models can be used to examine lead-lag relationships--helpful in assessing cause and
effect--of paleoclimate variables while addressing two other challenges that are often encountered in
paleoclimate data: unevenly spaced data; and switching between regimes at unknown times. Specifically,
we analyze multidimensional paleoclimate proxies, sea surface temperature (*SST*), $C_{37}$, $\partial^{15}N$, and %N
from the central Peru margin to find their correlations and changes in their variability over the Holocene
epoch. The four proxies are sampled at high-resolution but are not synchronously sampled at all possible
locations. The multidimensional records are treated as evenly spaced data with missing parts, and the
missing values are filled by the Kalman filter expected values. We employ hidden Markov models
(HMM) and autoregressive HMM (AR-HMM) to address the potential that the degree of variability and
the correlations between in these proxies appears to show changes over time. The HMM, which is not
autoregressive, shows instantaneous correlations between observables in two regimes. However, our
investigation of lead-lag relationships using the AR-HMM shows that the cross-correlations do not
indicate a causal link. Each of the four proxies has predictability on decadal timescales, but none of the
proxies is a good predictor of any other, so we hypothesize that a common unobserved variable--or a set
of variables--is driving the instantaneous relationships among these four proxies, revealing probable
confounding without prior knowledge of potential confounding variable(s). These findings suggest that
the variability at this site is remotely driven by processes such as those causing the Pacific Decadal
Oscillation, rather than locally driven by processes such as increased or decreased vertical mixing of
nutrients.
**Keywords:** Hidden Markov Model, Decadal Variability, Holocene, Pacific Decadal Oscillation,
Paleoclimate

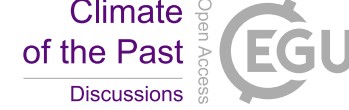



## 1.0 Introduction

This paper examines statistical aspects of a long-duration, high-resolution, multi-dimensional time series of four proxies ($SST$; $C_{37}$; $\partial^{15}N$; $\%N$) that record variations in marine conditions over the Holocene epoch (0.60 to 9.44 kA B.P.). The sediment is sampled at high-resolution to amount to roughly 3-year averages sampled every 7 years under the accumulation rate typical of the region. These records indicate both surface and subsurface variability in the physical and biological state. It is expected that the evolving relationships among these records over the Holocene reveal aspects of the mechanisms responsible for variabilities, such as correlations, timescales, and predictability.

The four records examined are proxies for sea surface temperature ($SST$) through the alkenone proxy, biological productivity of a specific phytoplankton group ($C_{37}$) through analyses of the abundance of alkenones (representing haptophyte algal productivity), subsurface properties through analyses of $\partial^{15}N$, an index of subsurface oxygenation and denitrification, and the percentage of organic nitrogen ($\%N$) which is a composite of all biological inputs to the sediment. Interannual and decadal variability is observed in subsurface oxygen fluxes and concentrations worldwide, but particularly in the eastern tropical South Pacific oxygen minimum zone where this core is located (Bopp et al. 2002, Stramma et al. 2008). These studies suggest that a combined examination of 1) warming of the ocean surface (here recorded through alkenone $SST$), 2) changes in stratification (here recorded through upwelling as indicated via productivity $C_{37}$ and $\%N$), 3) changes in ecological makeup (here recorded through a comparison between $\%N$ which indicates a combined productivity of all organisms and $C_{37}$ which indicates productivity of only some organisms), and 4) changes in the oxygen utilization at depth (here proxied through $\partial^{15}N$) may help explain the combination of thermal, dynamical, and biogeochemical factors contributing to the variability in this region. Sediment samples were placed on an age scale based on a polynomial fit to 8 radiocarbon dates - the resulting age model has an uncertainty on the order of 100 yr.

Some of the key questions in this region are whether the variability is from a local or internal source, such as variation in physics through mixing or eddies at the surface (Brink et al., 1983, Colas et al. 2012) or changes in the biological makeup of ecosystems in the region (e.g., Gooday et al. 2010), or from a remote or external source, such as variations in the water properties arriving at the site through large scale modes such as El Nino or the Pacific Decadal Oscillation (Mantua et al. 1997, Deser et al. 2010). The site (Fig. 1) is known for wind-driven upwelling (Brink et al. 1983) at depths shallower than 250m and low oxygen concentrations at depth typical of the eastern tropical South Atlantic oxygen minimum zone, which has been highly variable near 250m depth in recent times (Stramma et al. 2008). Despite the low oxygen levels at depth, the typical sediment accumulation rate over the Holocene during these samples is high (70 cm/kyr), which suggests high, sustained biological productivity and presumably a persistent level of oxygen demand.

A visual analysis of the proxy records (Fig. 2) suggests that the variability of four proxies might fall into multiple regimes: one state with high variability and another state with low variability. This *biphasic* behavior guided our initial analysis using a Hidden Markov Model (HMM; Rabiner, 1989). Hidden Markov methods are increasingly used in identifying climate regime shifts (e.g., Majda et al. 2006, Franzke & Woollings 2011, Ahn et al. 2017).

A less common tool in climate modeling is the autoregressive hidden Markov method (AR-HMM,
Hamilton, 1988, 1989, 1994) which allows for some memory in the system through a dependence on
previous proxy values as well as correlations in the present proxy value noise. Both our HMM and AR-
HMM results show that there exist two regimes of variability in proxy space at site MW8708-PC2. Here
the AR-HMM technique will be used to probe deeper into distinctions between causality and correlation,
under the premise that a predictive cause should precede its effect in time. A surprising result of this study
is that our conception of the relationships among these proxies changed dramatically when this technique
was applied and contrasted to the more standard HMM approach. The AR-HMM shows that both regimes
show high auto-correlation and low cross-correlation, thereby indicating that none of the proxies are good
predictors of other proxies on interannual timescales. In cases in which regime change is not present, a
simpler autoregressive only model will be sufficient to assess predictive cause. The software provided
with this paper (https://github.com/seonminahn/ARHMM) can be applied for the analysis of multi proxy
data from a core record. When inferences on predictive cause between cores is of interest, it is essential to
account for uncertainty in age estimates to ascertain the significance to a putative lead/lag
**1.1 Context from Modern Observations**
To better understand what processes would affect the variability on the timescales that are sampled, a
brief analysis of the region and related climate indices was carried out. The location is somewhat south of
the region most active during the El Nino/Southern Oscillation (ENSO) cycles (i.e., south of NINO1,
Rasmussen & Carpenter, 1982). On longer timescales, a meridionally broader, yet similarly shaped
pattern of variability has come to be known as the Pacific Decadal Oscillation (PDO) (Deser et al. 2010).
Fig. 1(a) shows the location of the MW8708-PC2 sediment core that is analyzed for this study. The
location is superimposed on a map of the correlation of global sea surface temperature with the nearest
HadISST data point (15.5S, 75.5W). Warming in this region correlates well with warming along the
central and eastern equatorial Pacific, cooling over most of the extratropical Pacific, and weakly
correlates with temperatures in other basins. The correlated pattern resembles both the El Nino pattern
and the Pacific Decadal Oscillation pattern (Deser et al. 2010). In time, the sediment record (indicated by
circles in Fig.1b) is too infrequent to capture El Nino variability, generally taken to dominate the 2-7 year
band. While some of the biggest El Nino events (1982-3, 1997-8) are still visible in the filtered data, it is
evident that time filtering similar to our sediment sampling has removed most of the high-frequency
ENSO variability.
Deser et al. (2010) follow Mantua et al. (1997) in tracking the Pacific Decadal Oscillation using the first
empirical orthogonal function (EOF) and principal component (PC) of North Pacific sea surface
temperature (20N to 70N, 100E to 100W) after the removal of seasonal and global mean variability. Its
variability is taken as a PDO index. This index captures much of the low frequency (>7yr) variability near
the sediment location (Fig. 1b), even though the core location is remote from all data included in the PDO
index and the monthly index and core location SST have a correlation coefficient of only 0.16. Running
means over 2 to 22-years of SST at our site correlates with the PDO index all have correlation coefficients
above 0.53, and the peak coefficient is just above 0.6 (for 7-year averages). Thus, we interpret the
dominant mode of variability accurately sampled by the core measurements to be associated the Pacific
Decadal Oscillation. Note that the record described here is significantly longer than extant records of the
PDO (e.g., AD 993-1996 tree-ring compilation by McDonald & Case, 2005).




A variety of mechanisms have been used to explain the PDO. Alexander (2010) reviews the mechanisms
and concludes that a variety of causes are consistent with the observations, mainly heat flux and wind
variability, including El Nino variability communicated to the N. Pacific by the "atmospheric bridge".
This variability is modulated toward lower frequencies by the reddening of "stochastic" variability
(Hasselmann, 1976) by the large heat capacity of the mixed layer (Frankignoul & Hasselmann, 1977), but
also through slow-response phenomena such as the re-emergence of sub-boundary-layer temperature
anomalies during subsequent winters and the slow propagation of baroclinic Rossby waves. The
autoregressive formulation of the AR-HMM is essentially the same as the stochastic model used by
Hasselmann. According to Frankignoul & Hasselmann (1977), forcing amplitude affects response
amplitude, but the damping rate of variability affects both the magnitude of variability and the persistence
timescale, with greater magnitude and longer persistence indicating weaker damping which is a
consequence of a shallower mixed layer and reduced heat capacity.

**2.0 Method**
**2.1 Data Collection**
High-resolution records of four paleoclimate indicators are collectively analyzed for a sediment core
retrieved from the central Peru margin (Site MW8708-PC2: 15.1°S, 75.7°W, water depth of 250m, Fig.
1). This site has an extremely high and steady sedimentation rate (70cm/kyr) across most of the Holocene
(10kA - 1.4 kA), and frequently contains annual laminations. Records are obtained from 2cm (3 years)
slices taken every 5cm (7 years). The age model determined the core top to be located at ~600 years
before present (bp), (gravity coring typically disturbs the upper few decimeters of sedimentation and the
base of the record to lie at ~9440 yr bp. The very gentle curvature in estimated sediment accumulation
rates (Chazen et al., 2009) will be ignored in this study, so depth is proportional to age and time steps are
uniform.

**2.2 Missing Data**
The four proxies are measured in high-resolution with fairly uniform depth sampling (2cm about every
5cm), but different proxies are not sampled at all possible locations. In order to compose an evenly-
spaced data set that will be used to train discrete-time statistical models described below, the expected
values in an evenly-spaced record are used to fill in the records using a Kalman filter (Little & Rubin,
1986; Viefers, 2011). The Kalman filter finds the expected value of the missing data given the observed
value, and we find the maximum likelihood estimates of the model parameters by using the expectation-
maximization algorithm. Before doing so, time from 0 to 563 discrete time steps (each of which
represents 5cm/7yr) is discretized into 1127 discrete half-time-steps (each of which represents
2.5cm/3.5yr, or approximately the width of an analysis slice). Each proxy analysis is then allocated to the
half-time-step nearest its location in depth/age. Not every possible slice was analyzed: there are 526 *SST;*
526 $C_{37}$; 727 $\partial^{15}N$; and 728 *%N* measurements out of 1127 possible to fill all half-time-steps.

Each half-time-step is interpreted as a 4-component vector of observations $X(t)$.
$$X(t) = \begin{bmatrix} x_1(t) \\ x_2(t) \\ x_3(t) \\ x_4(t) \end{bmatrix} = \begin{bmatrix} (SST(t) - \langle SST \rangle)/\sigma_{SST} \\ (C_{37}(t) - \langle C_{37} \rangle)/\sigma_{C_{37}} \\ (\delta^{15}N(t) - \langle \delta^{15}N \rangle)/\sigma_{\delta^{15}N} \\ (\%N(t) - \langle \%N \rangle)/\sigma_{\%N} \end{bmatrix}$$



In the climate and data assimilation literature, this vector is usually called the "state" vector; here it will
be called the observation vector to distinguish it from the regime or "state" of the hidden Markov model.
After arranging the data in this manner, the expected values estimated using a Kalman filter are used to
fill in missing data (Figs. 3, 4). The mean and standard deviation of each proxy variable have been
removed as a preprocessing step so that the different units of each measurement are not a factor and the
Kalman filter likewise does not depend on the units of measurement. Our preparation of this discrete-time
technique and the discrete-time statistical models below assume that even spacing in depth is sufficiently
uniform in time, i.e., variations in the age-depth relationship were not considered in this imputing
technique.
**2.3 Statistical Models: HMM and AR-HMM**
The degree of variability in correlation among these proxies appeared to change at unknown times over
this epoch. Visual analysis suggests that the correlations and variability of the four proxies varied over
time in a potentially abrupt manner (Figs. 2, 3, 4). Indeed, use of a two-state (a.k.a. two-regime) hidden
Markov models (HMM) and a generalization of this approach, autoregressive HMM (AR-HMM), do
detect two distinct states at this site, characterized by different levels of variability and predictability.
Experimentation with higher numbers of states revealed that two states were sufficient for this record.
Two-state hidden Markov models are considered using two different emission (time-correlation or
memory) models. The first model assumes conditional independence among observations given the state,
regime, and the second model considers direct dependence with adjacent observations (i.e., memory). The
first one is consistent with a general vector, or multivariate Hidden Markov Model (HMM), and the
second one is called the autoregressive hidden Markov model (AR-HMM), which is also known as a
switching autoregressive model (Hamilton, 1988, 1989, 1994). Both models have hidden regimes or
"states" in which it is assumed that the historical dependence of the current hidden (unobserved) state is
entirely accounted by the state of its immediate proceeding neighbor and a transition probability, i.e., the
state-switching process is Markovian. The matrix of state transition probabilities is
$$a_{ij} = P(s(t+1) = j \mid s(t) = i)$$
The difference between the AR-HMM and HMM models is the relationship between the observations at
different times. The equation for the AR-HMM can be written
$$X(t) = c_{s(t)} + \theta_{s(t)} X(t - \Delta t) + \epsilon(t)$$

$$\begin{bmatrix} X_1(t) \\ X_2(t) \\ X_3(t) \\ X_4(t) \end{bmatrix} = \begin{bmatrix} c_1 \\ c_2 \\ c_3 \\ c_4 \end{bmatrix} + \begin{bmatrix} \theta_{11} & \theta_{12} & \theta_{13} & \theta_{14} \\ \theta_{21} & \theta_{22} & \theta_{23} & \theta_{24} \\ \theta_{31} & \theta_{32} & \theta_{33} & \theta_{34} \\ \theta_{41} & \theta_{42} & \theta_{43} & \theta_{44} \end{bmatrix} \begin{bmatrix} X_1(t - \Delta t) \\ X_2(t - \Delta t) \\ X_3(t - \Delta t) \\ X_4(t - \Delta t) \end{bmatrix} + \begin{bmatrix} \epsilon_1(t) \\ \epsilon_2(t) \\ \epsilon_3(t) \\ \epsilon_4(t) \end{bmatrix} \tag{1}$$

There are two constant vectors $c_{s(t)}$, which are selected depending on the state at time $t$ Likewise, the
autocovariance regression matrix ($\theta_{s(t)}$) that prescribes the deterministic part of the model evolution
based on observations at a previous time and the noise covariance matrix ($\Sigma_{s(t)}$) that prescribes the
stochastic part of the model evolution also have two versions which are selected based on state. The noise
vector ($\epsilon(t)$) is chosen at each time from a Gaussian white noise distribution with zero mean and
covariance matrices $\Sigma_{s(t)}$ that contain all of the information about stochastic variances and covariance of
the observations.



The HMM can be written the same way as (1), but removing the deterministic dependence of current
observations on previous observations ($\theta = 0$). The HMM assumes that each observation follows
multivariate normal distribution with means ($c$), (stochastic) variances ($\Sigma_{ii}$), and (stochastic) covariances
($\Sigma_{ij}$) determined only by present value of the hidden state.
In both the HMM and AR-HMM models, the unknown parameters including constant parameters in each
two-state model are estimated by the Baum-Welch expectation maximization algorithm (EM; Rabiner,
219    1989).
**3.0 Parameter Estimation Results**
The parameter estimations are done using the EM algorithm for both HMM and AR-HMM. The EM
algorithm updates parameters iteratively using the forward and backward sampling algorithm. The data
augmentation step that uses the Kalman filter is added at the beginning of each iteration to address
missing data. Depending on initial conditions of the EM algorithm, it is possible that the EM algorithm
converges to local maximum estimators instead of global maximum estimators. To avoid local maxima,
parameter estimations are repeated with 100 different initial conditions and the selected parameters are
those that achieve maximum likelihood from this set.
**3.1 HMM Parameters**
For a two-state HMM after removal of the overall mean and normalization of the standard deviation of
each proxy, there are five unknown parameters which have 32 degrees of freedom in total: the transition
matrix $a$, and one version of $c$ and $\Sigma$ for each state. Table 1 shows the results of the parameter
estimations. The two states are distinctively different in means and covariance. The mean of each proxy
differs in sign between the two states, which must be the case as the overall mean of each proxy has been
removed. However, the pattern of means among the proxies, e.g., high $SST$ and low $C_{37}$, is a signature of
each state. The absolute values of the components and eigenvalues of $\Sigma$ are larger in state 1 than in state
2. The eigenvalues (the strength of correlated noise components) of $\Sigma$ are 2.21, 0.98, 0.57, and 0.18 for
state 1 and 0.82, 0.58, 0.20, and 0.13 for state 2. Thus, we can associate state 1 as a "noisy" state and state
2 as a "calm" state, because the proxies tend to fluctuate more when in state 1 than in state 2. In terms of
transition probability, the diagonal elements of $a$ are close to 1, which implies that there is a high
probability of staying in a state. Table 2 shows that 7 of the 12 correlation coefficients are approximately
0.5 or higher.
According to the parameter estimations, the most probable state is determined at each time using the
backward sampling (Fig. 3). The median (mean) time to remain in HMM state 1 over 1000 samples is 70
years (128.6 years). The median time to remain in state 2 over 1000 samples is 91 years (189.8 years).
**3.2 AR-HMM Parameters**
For a two-state AR-HMM after removal of the overall mean and normalization of the standard deviation
of each proxy, there are seven unknown parameters which have 64 degrees of freedom in total: the
transition matrix $a$, and one version of $c$, $\theta$, and $\Sigma$ for each state. The estimated parameters are shown in
Table 2, and the model state and imputed values are shown in Fig. 4. Again, state 1 can be identified as
the "noisy" state and state 2 is "calm". In terms of transition probability, the diagonal components of $a$ are
around 0.8, which are smaller than those of the HMM. Thus, there are more frequent state changes in Fig.



4 than shown by the HMM (Fig. 3). The median (mean) time to remain in state 1 over 1000 samples is 7
years (17.9 years). The median time to remain in state 2 over 1000 samples is 28 years (39.8 years).
The diagonal entries of $\theta$ are close to 1 on both states: each variable of state 2 depends strongly on its
own past value. The diagonal entries of $\theta$ for state 1 are smaller than that of state 2 with both greater than
0.85 for all four proxies. The off-diagonal entries are all smaller than 0.07 for both matrices. Thus, only a
small part of the dependence of each variable on its past value can be attributed to cross-correlations
rather than autocorrelations. The antisymmetric components of $\theta$ are much smaller than the diagonal
components, so the "probability angular momentum" which lends covariant predictability (Weiss et al.
2016, Zia et al. 2016) is not significant.
The diagonal entries of $\Sigma$ in the AR-HMM are much smaller than they were in the HMM--so that
variability attributed to noise within each variable is considerably lessened by the introduction of
memory. The eigenvalues of the $\Sigma$ matrix as well are roughly a factor of 5 to 50 smaller, indicating that
the covariant modes of noise are estimated to be much weaker when the memory of the AR-HMM system
is permitted.
The mean state $c$ of the HMM and AR-HMM do not resemble one another in its pattern, magnitude or
sign. Thus, while these patterns are a characteristic of the HMM and AR-HMM states, there is no
agreement between the pairs of states in mean, timing of onset, or cross-correlations.

**3.3 Comparison of Models**
The HMM is a special case of the AR-HMM. As the HMM may be formed from the AR-HMM, the fact
that the AR-HMM does not resemble the HMM implies that the lagged time information is a critical
aspect of the data. Thus, a key conclusion from the statistical models is that the lagged autocorrelations
are significantly better predictors of proxy variability than the different proxy-to-proxy cross-correlation
either at lagged times or as induced by correlated noise (Fig. 5). This fact implies that the different
proxies are not causally related to one another, as is often assumed in multi-proxy paleoclimate analyses
(Hu et al. 2017). Thus, in this location, the four proxies (*SST; $C_{37}$; $\partial^{15}N$; %N*) are not related to each other
in the local sense that variability in any one dominates or contributes significantly to variability in another
through a local physical or biological mechanism.

For reference, the mean and variance of each proxy are given for noisy state (state 1) and calm state (state
2) of the HMM and AR-HMM in Tables 1 and 5. While both AR-HMM and HMM attribute a noisy state
and a calm state to the time series, none of the means, variances, or timing of onset of these states agree.
Furthermore, it was noted that the HMM mean states must be opposite in sign in order for the normalized
time series to be zero. The AR-HMM is not constrained by this limit, as the predictions of $\theta$ can contribute
to the mean. Because the AR-HMM is more general than the HMM, disagreement between these state
identifications indicates that the autoregression memory of the AR-HMM is important. Bolstering this
idea is the fact that the dominant modes of correlation of observations with the previous time observations
are autocorrelations, i.e., the dominant predictor of any of the four proxies is itself at a previous time and
not interactions between the observed variables.





For the comparison with the AR-HMM, the correlations of the four proxies in HMM are estimated as in
Tables 3-4. These correlation matrices are calculated using each data set in which the missing parts have
been imputed by their expected value and the state estimation at each time. The signs of correlations are
usually the same between the two model assessments, but the strength of the cross-correlations vary
somewhat. Note that the cross-correlations do not disappear in the AR-HMM. Even though the full model
reveals the underlying autocorrelations, these simple single-time correlations are unable to detect any
inconsistencies that correlations between variables do not reveal causation between variables in this data.

**4.0 Discussion**
The preceding statistical model results may be related back to the original science questions that
motivated this collection of data. That is, what changes in physics or biological makeup helps better
understand the mechanisms at play in setting the variability in this region?

**4.1 Implications for Mechanisms**
In the introduction, it was argued that potential local mechanisms might be used as causes to explain
correlations and connections among these data. Variability in upwelling, stratification, biological makeup,
oxygen utilization and productivity, and many other mechanisms would be likely to strengthen a
particular set of cross-correlations and levels of variability among these data. Indeed, two different states,
one noisy and calm, were detected with both AR-HMM and HMM model parameter estimation. Tables 1,
3, 4, and 5 show significant cross-correlations and difference in cross-correlations and levels of variability
between these two states. The typical HMM approach confirmed roughly these conclusions.

However, a closer examination of the dependences of the proxies on AR-HMM autocorrelations with
their previous time values and cross-correlations with previous and synchronous values of other proxies
reveals a very different story. This analysis revealed that the restrictions required to reduce the AR-HMM
to the HMM, i.e., the neglect of memory of past observations, systematically corrupted interpretation of
the system. The magnitude of the components and eigenvalues of the $\Sigma$ matrix are significantly smaller in
the AR-HMM than in the HMM. Thus, present observations are caused--in the Granger (1969) sense--by
the previous observations, i.e. the predictive rather than the intervention sense. The small off-diagonal
terms in $\theta$ indicate that each proxy is not strongly caused by any other proxy, only by its own previous
values. Rather, the apparent correlations found by the HMM model very likely stem from confounding
(https://explorable.com/confounding-variables) by an unobserved mechanism that drives all four
parameters in a coordinated manner. These results are inconsistent with any local mechanism that would
link these proxies to one another causally, e.g., if $SST$ variability were to indicate upwelling that drives
productivity and thus $C_{37}$ and $\%N$. Because both the past-time cross-correlations and the present-time
correlated noise became less consistent in the AR-HMM when compared to the HMM, it is unlikely that
this lack of cross-predictability is due to the limited temporal resolution. Consistent local mechanisms
would require variability caused by unobserved mechanisms that might affect one or more of the proxies,
so-called confounding variables. A variety of distinct remote causes for variability, e.g., $SST$ driven by the
PDO and other proxies driven by other climate modes or source variability, are a sufficient explanation
for the results here.

**4.2 Implications for Predictability**



One interesting aspect of the AR-HMM model is that it reveals the dependence of the present
observations on previous observations. This implies a sort of predictability of the four proxies based on
the AR-HMM. However, because the predictability is essentially just autocorrelations, the AR-HMM
does not predict significantly differently from persistence (same observations next time as this time).
Nonetheless, some aspects of predictability in this system are of interest.

One difference between a prediction system and a reanalysis of past events is that a prediction system
should use only the data that precedes the times that will be predicted. Two methods to achieve this were
used here: 1) predict new parameters using the data sequence preceding the points we predict, and 2)
sample values using these parameters.

Predictability of the AR-HMM was evaluated over two time windows: 236-266 and 535-563 cm depth.
Fig. 6 gives a sense of what behaviors these predictions tend toward in the 236-266 window. The interval
236-266 is chosen because the resolution of the interval 236-266 is relatively higher than other intervals,
and the AR-HMM state is persistently in the (calm) state 2 over this interval. Taking 266 as an endpoint,
the predictability of one-step to thirty-step is assessed. The interval 535-563 includes the most recent data
and tends to remain in the (noisy) state 1. Each prediction is repeated 1000 times.

Depending on the most probable state of an initial point, the entries of the next step are computed with the
emission model (equation (2)) with parameters estimated in the previous section. The state of the next
step is determined by the transition probability, and then the entries of the following step are computed
with the equation (2) in the same way. State determination and entry computations are repeated until
reaching the endpoint.

The accuracy of predictability based on the AR-HMM is examined using mean squared errors (MSE).
Predictions up to four-step, which corresponds to approximately three decades, achieve reduction of the
MSE by 40-80%, depending on the proxy. The results do not show a tight range of prediction when the
length of prediction is longer than four steps ahead. However, the probability of remaining in a given state
or regime for the future steps can be predicted from the transition probability, typically for decades based
on the AR-HMM transition probabilities. The noisiest proxies tend to have forecasts that revert to
spanning their climatological range most quickly. The forecasts that begin in the noisy regime of state 1
tend to lose persistence faster as well.

In order to compare the HMM with the AR-HMM, we assessed the predictability of the HMM is assessed
in the same manner as the that of AR-HMM. While the MSEs increase as the forecast length increases in
AR-HMM predictions, the MSEs of HMM keep the same size regardless of prediction length. In a system
with strong auto-correlations such as this one, useful forecasts require a memory of past states.

**5.0 Conclusions**
Multi-proxy records are a potentially powerful tool in strengthening understanding of paleorecords.
However, depending on which variables are observed and where, they may or may not capture direct
evidence of the mechanisms at work. This study was carefully designed to distinguish different types of
local mechanisms that might be causing variability on the Peru margin over the Holocene. However, it is



our interpretation of the estimates of statistical model parameters found that no local causal mechanisms
were observed to be significant at the roughly decadal scale of sampling employed.

This study illustrates the importance of assessing predictive (Granger) causation in order to avoid
spurious diagnoses of the mechanism through the use of autoregressive (AR) models for example. AR
algorithms are widely available (in R and MATLAB) for cases not involving regime change. In addition
as pointed out by Hu et al. (2017), when multiple records are involved, age uncertainty can also lead to
spurious associations.

Before closing, it is interesting to consider broadly the implications of the regime-switching observed
here. While it was shown that similar-sampling-frequency analyses of modern observations at this
location reveal SST variability that is dominated by the PDO, past variability indicates a change in PDO
variability at this site, transient appearance of other dominant modes, or changes in teleconnections.
Stevenson et al. (2012) demonstrate that changes in such remote influences of climate variability are
likely to be common even when the underlying climate mode is unchanging.


**Funding**
BFK was supported by NSF 1245944.

**Author Contributions**
Conceptualization, T.W. and C.E.L; Methodology, S.A. and C.E.L; Software, S.A.; Formal Analysis,
S.A., B.F.-K., T.W., and C.E.L; Data Curation, T.H; Writing, S.A., B.F.-K., T.W., and C.E.L;
Visualization S.A.

**Acknowledgements**
Conversations with Steve Clemens, Warren Prell, Jim Russell, Jeff Weiss, and Deborah Khider greatly
improved this work.

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



## Figures

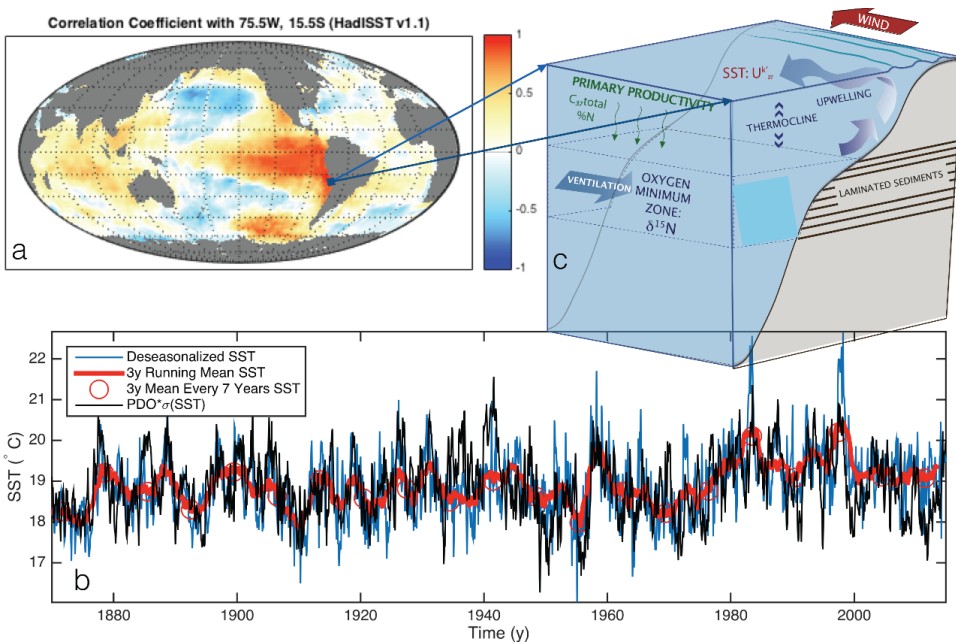

Figure 1 a) Location of the site MW8708-PC2 (15.1°S, 75.7°W, water depth of 250m), superimposed on
the correlation of the SST gridpoint nearest that location with each SST gridpoint globally (using
HadISST data, Rayner et al. 2003). b) Time series of sea surface temperature with climatological 1900-
1914 seasonal cycle removed (blue), 3-year running mean of this SST (red), Pacific Decadal Oscillation
principle component time series (Mantua et al. 1997, Deser et al. 2010) which has been rescaled to have
the same variance as the SST (black). The red circles are exemplars of 3-year averages plotted every
seventh year. c) A schematic of the region, illustrating the proxies examined ($SST; C_{37}; \partial^{15}N; \%N$) and
local physical processes (wind-driven upwelling, thermocline, oxygen minimum zone).





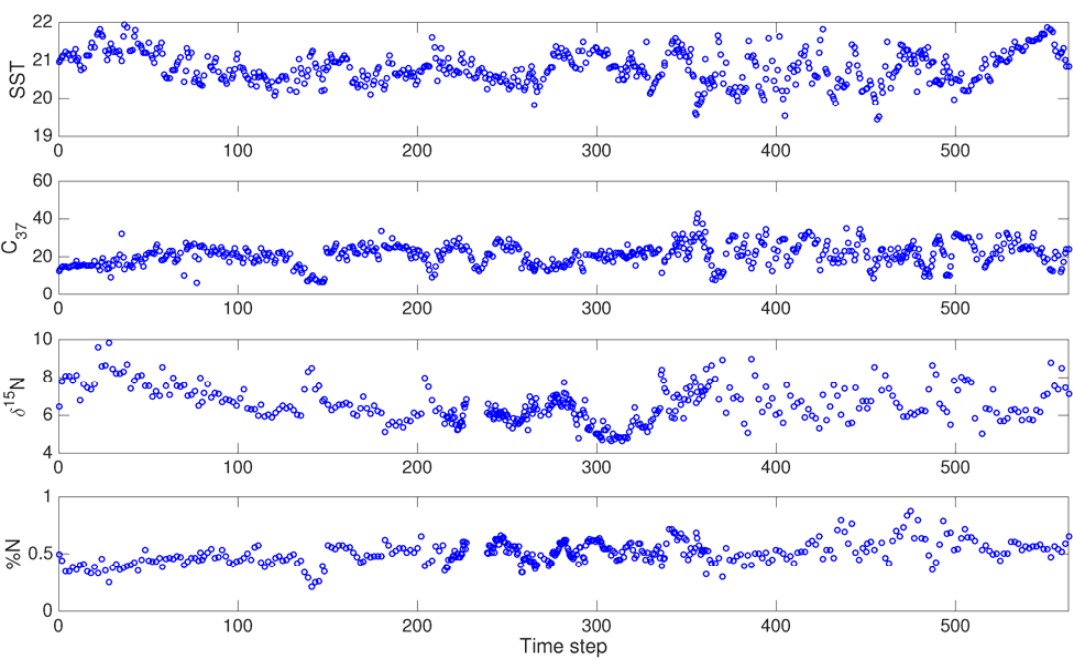


Figure 2 Observed data for time steps 0 to 563 (0.60 to 9.44 kA B.P.), with being the most recent point
(time increasing to the right). 47% *SST* and $C_{37}$ are missing, and 65% of $\partial^{15}N$ and *%N* are missing.





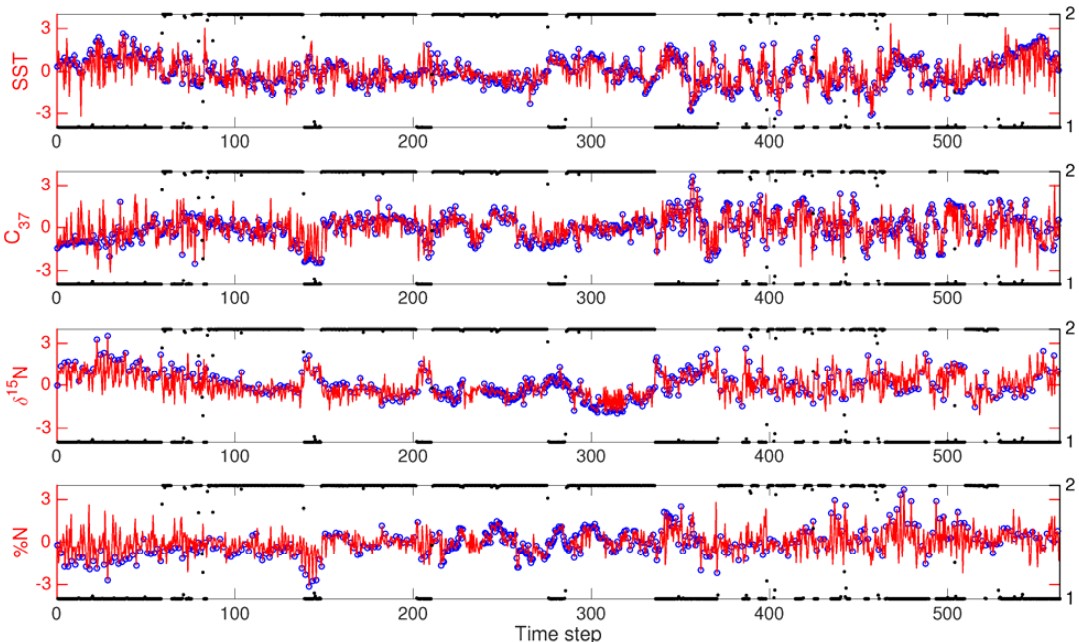

Figure 3 [HMM] State assignments by the HMM (black dots). State 1 is indicated by a black dot near the lower side of each graph, indicating the probability of being in the noisy state. State 2 is indicated by a black dot near the upper side of the graph. Indeterminate states are indicated by black dots in the middle of the graph. Also shown are observations (blue circles) whose missing parts are imputed by expectation values from the Kalman filter (red lines): *SST*, $C_{37}$, $\partial^{15}N$, and *%N* (from top to bottom).





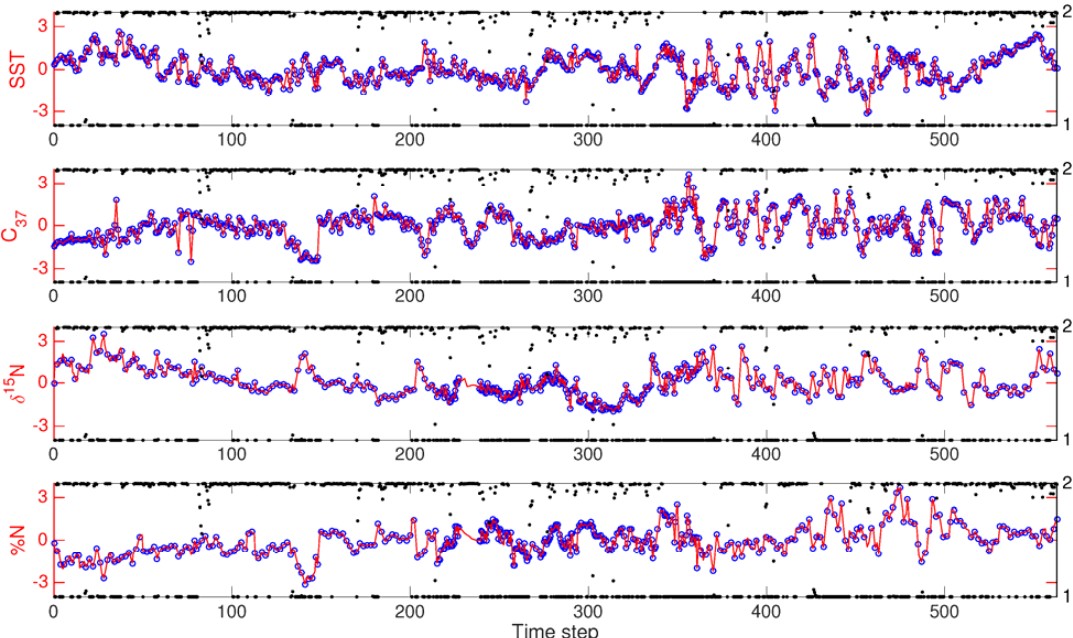

Figure 4 [AR-HMM] State assignments by the HMM (black dots). State 1 is indicated by a black dot near
the lower side of each graph, indicating the probability of being in the noisy state. State 2 is indicated by a
black dot near the upper side of the graph. Indeterminate states are indicated by black dots in the middle
of the graph. Also shown are observations (blue circles) whose missing parts are imputed by expectation
values from the Kalman filter (red lines): *SST, C$_{37}$, $\partial^{15}N$, and %N* (from top to bottom).





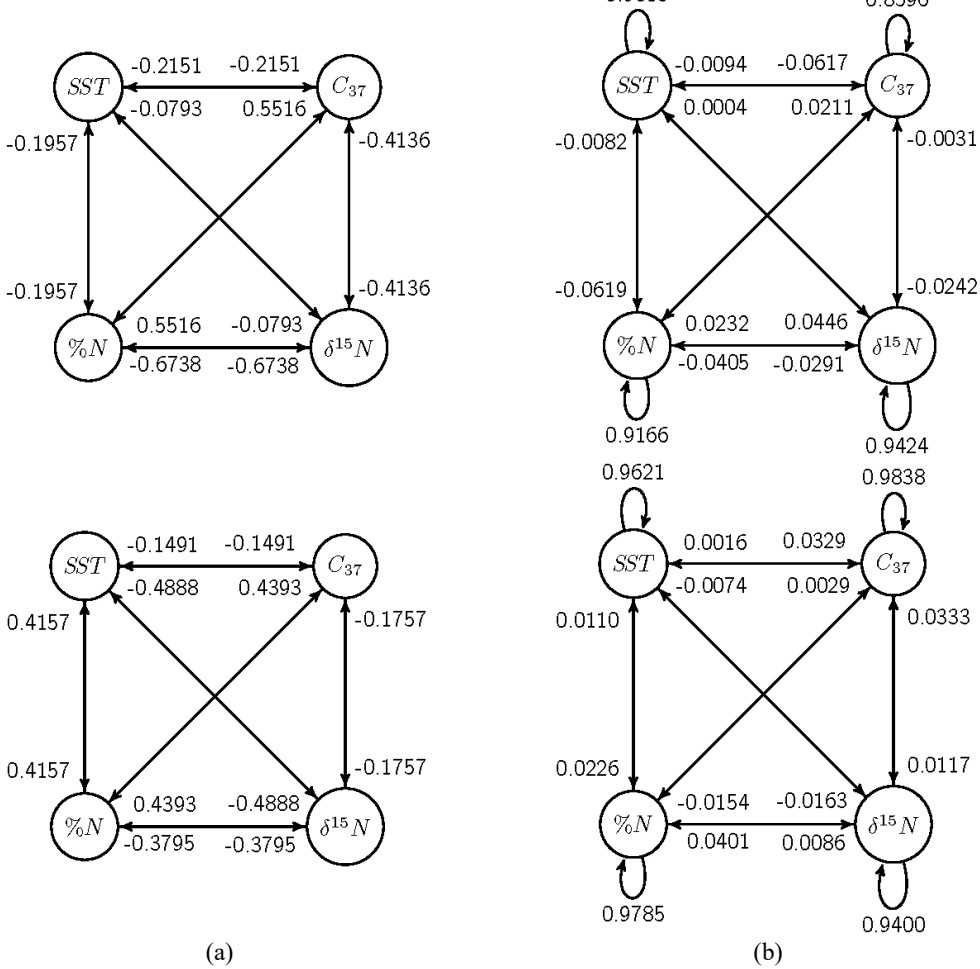

(a) (b)

Figure 5 Dependencies among observations and hidden states through a visual schematic of the
correlation matrices for (a) HMM and (b) AR-HMM. Nodes are connected with an arrow if one node at
the head of an arrow depends on another node at the origin of an arrow. The loopback dependencies in b)
indicate a correlation of the present state of that variable with its value at a previous time.



Figure 6 [AR-HMM] Results of 4-step prediction from t=264 to 266. The multiple grey lines indicate
1000 individual forecasts that differ in noise and state transitions. The black errorbars indicate the 1000-
forecast 0.05 quantile and 0.95 quantile, and the red circles indicate the observed values. The red dotted
lines indicate the range of the observation data and the solid red lines show the 0.05 quantile to 0.95
quantile of the observed data.





**Tables**
Table 1 [HMM] Parameters estimated for the HMM.

| | | State 1 (Noisy) | | | | State 2 (Calm) | | | |
|---|---|---|---|---|---|---|---|---|---|
| | | $SST$ | $C_{37}$ | $\delta^{15}N$ | $\%N$ | $SST$ | $C_{37}$ | $\delta^{15}N$ | $\%N$ |
| $c$ | | [ 0.5044 | −0.1731 | 0.7302 | 0.0602 ] | [ −0.4007 | 0.1769 | −0.5051 | −0.0728 ] |
| $\Sigma$ | | 0.9962 | −0.2371 | −0.0818 | −0.1988 | 0.5076 | −0.0479 | −0.2130 | 0.1769 |
| | | −0.2371 | 1.4041 | −0.3311 | 0.7157 | −0.0479 | 0.5212 | −0.0875 | 0.2075 |
| | | −0.0818 | −0.3311 | 0.4766 | −0.4672 | −0.2130 | −0.0875 | 0.3323 | −0.1307 |
| | | −0.1988 | 0.7157 | −0.4672 | 1.0716 | 0.1769 | 0.2075 | −0.1307 | 0.3702 |

Transition Probability
$$\begin{bmatrix} 0.9555 & 0.0445 \\ 0.0341 & 0.9659 \end{bmatrix}$$

Table 2 [AR-HMM] Parameters estimated for the AR-HMM.

| | | State 1 (Noisy) | | | | State 2 (Calm) | | | |
|---|---|---|---|---|---|---|---|---|---|
| | | $SST$ | $C_{37}$ | $\delta^{15}N$ | $\%N$ | $SST$ | $C_{37}$ | $\delta^{15}N$ | $\%N$ |
| $c$ | | [ 0.0211 | −0.0001 | 0.0557 | −0.0067 ] | [ −0.0185 | −0.0012 | −0.0358 | 0.0124 ] |
| $\theta$ | | 0.9083 | −0.0094 | 0.0004 | −0.0082 | 0.9621 | 0.0016 | −0.0074 | 0.0110 |
| | | −0.0617 | 0.8596 | −0.0031 | 0.0211 | 0.0329 | 0.9838 | 0.0333 | 0.0029 |
| | | 0.0446 | −0.0242 | 0.9424 | −0.0291 | −0.0163 | 0.0117 | 0.9400 | 0.0086 |
| | | −0.0619 | 0.0232 | −0.0405 | 0.9166 | 0.0226 | −0.0154 | 0.0401 | 0.9785 |
| $\Sigma$ | | 0.2157 | −0.0250 | −0.0080 | 0.0361 | 0.0333 | −0.0037 | 0.0105 | −0.0012 |
| | | −0.0250 | 0.2430 | −0.0692 | 0.0616 | −0.0037 | 0.0334 | 0.0022 | 0.0053 |
| | | −0.0080 | −0.0692 | 0.1644 | −0.0397 | 0.0105 | 0.0022 | 0.0063 | −0.0013 |
| | | 0.0361 | 0.0616 | −0.0397 | 0.2066 | −0.0012 | 0.0053 | −0.0013 | 0.0063 |

Transition Probability
$$\begin{bmatrix} 0.8077 & 0.1923 \\ 0.1720 & 0.8280 \end{bmatrix}$$

Table 3 [HMM] Correlation matrix of $SST$, $C_{37}$, $\partial^{15}N$, and $\%N$ for each HMM state. The correlation
matrices are obtained directly from the data set augmented by their expected values once the state at each
time is known.

| State 1 (Noisy) | | | | State 2 (Calm) | | | |
|---|---|---|---|---|---|---|---|
| 1.0000 | −0.2151 | −0.0793 | −0.1957 | 1.0000 | −0.1491 | −0.4888 | 0.4157 |
| −0.2151 | 1.0000 | −0.4136 | 0.5516 | −0.1491 | 1.0000 | −0.1757 | 0.4393 |
| −0.0793 | −0.4136 | 1.0000 | −0.6738 | −0.4888 | −0.1757 | 1.0000 | −0.3795 |
| −0.1957 | 0.5516 | −0.6738 | 1.0000 | 0.4157 | 0.4393 | −0.3795 | 1.0000 |







Table 4 [AR-HMM] Correlation matrix of *SST, C₃₇, ∂¹⁵N,* and *%N* for each AR-HMM state. The
correlation matrices are obtained directly from the data set augmented by the Kalman filter imputed
values once the state at each time is known.

|  | State 1 (Noisy) |  |  |  | State 2 (Calm) |  |  |
|---|---|---|---|---|---|---|---|
| 1.0000 | −0.2139 | 0.1081 | −0.0145 | 1.0000 | −0.2555 | 0.2065 | −0.0175 |
| −0.2139 | 1.0000 | −0.3406 | 0.4048 | −0.2555 | 1.0000 | −0.2853 | 0.3684 |
| 0.1081 | −0.3406 | 1.0000 | −0.4365 | 0.2065 | −0.2853 | 1.0000 | −0.3714 |
| −0.0145 | 0.4048 | −0.4365 | 1.0000 | −0.0175 | 0.3684 | −0.3714 | 1.0000 |


Table 5 [AR-HMM] Squared bias, variance, and MSE of the prediction up to 266. (The numbers in
parenthesis represent the percentage over the longest prediction.)

| Length of Prediction |  | 1 | 2 | 4 | 30 |
|---|---|---|---|---|---|
| $SST$ | Bias$^2$ | 0.0305 | 0.1005 | 0.0919 | 0.1004 |
|  | Variance | 0.0237 | 0.0540 | 0.0404 | 0.1386 |
|  | MSE | 0.0541 | 0.1545 | 0.1323 | 0.2390 |
|  |  | (22.66) | (64.64) | (55.34) | (100) |
| $C_{37}$ | Bias$^2$ | 2.0373 | 2.6533 | 3.2239 | 68.0915 |
|  | Variance | 4.9198 | 9.6830 | 15.2824 | 26.5790 |
|  | MSE | 6.9571 | 12.3363 | 18.5064 | 94.6705 |
|  |  | (7.35) | (13.03) | (19.55) | (100) |
| $\delta^{15}N$ | Bias$^2$ | 0.0691 | 0.1700 | 0.3130 | 0.4763 |
|  | Variance | 0.0849 | 0.1996 | 0.3026 | 0.5750 |
|  | MSE | 0.1540 | 0.3695 | 0.6155 | 1.0513 |
|  |  | (14.65) | (35.15) | (58.55) | (100) |
| $\%N$ | Bias$^2$ | 0.0002 | 0.0003 | 0.0083 | 0.0154 |
|  | Variance | 0.0011 | 0.0018 | 0.0027 | 0.0053 |
|  | MSE | 0.0012 | 0.0021 | 0.0110 | 0.0207 |
|  |  | (5.97) | (10.20) | (53.41) | (100) |
