# Peer review of "Autoregressive Statistical Modeling of a Peru Margin Multi-Proxy Holocene Record Shows 2 Correlation Not Causation, Flickering Regimes and Persistence 3 4 Seonmin Ahn1, Baylor Fox-Kemper2, Timothy Herbert2, Charles Lawrence1 5 6 1Division"

_Climate of the Past, 2018_

## Referee Comment (RC1) · Anonymous Referee #1 · 1 Mar 2018

The article by Ahn et al. use statistical approaches applied to an exceptionally well resolved Holocene record of hydrological proxies from a marine core collected along the peruvian margin.

As I am a paleoceanographer, not a statistician, so I judge the proxy- and sedimentology-related processes with only a limited scope on the statistical methods.

First, I could observe the efforts made by the authors to explain as much as possible the terms and concepts used in their analysis, which are helpful to the marine geologist reading the article to figure out what is meant. But this effort is far from being enough, though, if the authors' aim is to advertise the benefits that such analysis could provide to geologists who are interested in the method to their own records. In the text, the authors stack statistical concepts, description of numbers located in complex matrix, one on top of each other, and the geologist gets confused. I then urge the authors to wrap-up the article in a more comprehensive way. It is very frustrating, after reading a sentence such as ''Thus, a key conclusion from the statistical models is that the lagged autocorrelations are significantly better predictors of proxy variability than the different proxy-to-proxy cross-correlation either at lagged times or as induced by correlated noise (Fig. 5).", to actually have a look at Figure 5 that is otherwise quite confusing. As it stands, the article is not written appropriately to be informative to the general audience of the journal ''Climate of the Past".

This being said, the claims put forward in the article are interesting, and intriguing, but again the statistical analysis is far too disconnected from the sedimentological and proxy-related processes to be of interest for geologists. More discussion about what you're really dealing with is here warranted.

For example, it is notorious that alkenone-based SST estimates along the peruvian margin are warmer than the mean-annual SST if global core-top calibrations are applied (see e.g. Prahl, 2010, GCA; Kienast, 2012, Paleoceanography), probably because coccolithophorids live over time periods when upwelling ceases (e.g. the summer-stratified season of during El Niños). It implies that, in the global view that intensified upwelling is expected to increase productivity and decreases SST, the up-welling intensity / SST relationship you suggest might not intuitively be correlated as the authors claim, since alkenone-based SST might not be a good predictor of upwelling-related SST decreases. In the same vein, alkenone concentration might not be a good predictor of upwelling-induced productivity, since they might be synthesized during more stratified periods. I wonder why, in the end, you don't use biogenic opal instead of alkenone concentration to infer upwelling-induced productivity. The Chazen paper

shows there is quite a different signal in bio. silica compared to alkenone concentrations.

Also, you may want to comment more on the long-term evolution of your statistical outputs. If for some reason there is a long-term factor that strengthens productivity from the mid to the late Holocene, it is expected that oxygen consumption drove the appearance of laminations in the late Holocene, which artificially enhances variability (through decreasing bioturbation). Where exactly laminations occur? How bioturbation can act as a filter which would heavily lead your AR-HMM model to spuriously outperform the decadal predictability? It is easy to imagine that your memory effect could almost exclusively be driven by sediment mixing, and more discussion on this artifact must be discussed. In your figure 4, I visually tend to see the black dots trending from increasing densities from the ''calm''Âăto the ''noisy state'', which could be the signature of an increase in the occurrence of laminations through time.

To conclude, I felt the authors completely forgot the object they're looking at: an amazingly well resolved dataset that is amazingly complex to interpret because of the uncertainties associated with the proxies employed. The authors claim that their statistical, once applied blindly to a dematerialized set of numbers, ''showsÂăcorrelation not causation, flickering regimes and persistence'' (see the title). I suggest the authors to pause and think more about what is actually measured, and how the signal goes through sedimentary processes prior to fossilization in the geological record.

---

## Referee Comment (RC2) · Anonymous Referee #2 · 30 Apr 2018

The article aims at presenting a new way of investigating the causal relationship between different proxies extracted from the same core through the analysis of conditional probabilities in an autoregressive model with two states. I particularly enjoyed that the authors made their software public on Github.
I found that the article was quite "technical" and the journal "Climate of the Past" is probably not a convenient choice. Despite the fact that the approach is original and potentially useful to paleoceanographers, I think the authors should review the article and go deeper into the statistical analyses before resubmitting the manuscript. I also

suggest the authors to think about which journal to (re)submit in. I would have opted for a journal like "Nonlinear processes in geophysics".

Please find below the main points which I think should be improved. As I am a theoretical climatologist, I focus here on the methodology.

1. Basic correlation analysis.
   As it is mentioned in the title and the abstract, the study aims at showing that this new method is better than basic correlations when we want to explain causal relationships. I suggest to add a full paragraph where you show the classical correlation analysis and point out its main disadvantages, in order to better motivate the subsequent techniques.

2. In the introduction, you should explicitly state that the four proxies come from the same core, as it is an important implicit hypothesis.

3. In the introduction, the choice for the age model should be better discussed. In particular, you should discuss the changes you expect if you work with state-of-the-art age models based on Bayesian statistics, from a the technical point of view (is it technically feasible to fully consider the probability densities in your framework?) and from a climatological point of view (is it worthwhile?).

4. In section 2.3, you mention that considering more than two states basically gives the same results. I suggest to show it in an appendix. Moreover, you should first study the one-state case and show its limitations.

5. On the model.
   I am quite familiar with such equations but it took me a lot of time to understand your model. You should better explain that it is the same equation for the two states but with different parameters. Writing explicitly the HMM case would also make things clear. As the HMM is not well-known in the climate community, I

suggest you to briefly review the technique in an appendix jointly with the Baum-Welch algorithm, and in particular the concept of "hidden variable".

For a given state, the model is basically a 4-D Gaussian white noise (HMM) or a 4-D Gaussian red noise or AR-1 (AR-HMM). This should be explicitly mentioned. It turns out that the estimation of the coefficients of such processes can be performed from irregularly sampled time series, i.e. it does not require the data to be interpolated, as you do through the Kalman filter. See e.g. Gardiner, 2009, Stochastic methods, 4th edition, Sect. 4.5.6, and Kelly et al., 2014, the astrophysical journal, 788: 33. Transferring these techniques to your case is certainly not direct, but I think it is worthwhile investigating it to get rid of the interpolation procedure, which may cause uncontrolled biases in the analysis, especially if you want other people to use your code on data with any possible sampling schemes.

6. In section 3.3, you conclude that the four proxies are not causally related. I think you should better motivate this by a proper statistical test rather than just looking at the numbers in the tables presented at the end of the article.

7. In Fig. 2, 3 and 4, please draw in function of the time/age rather than the time step.

8. In Fig. 5, you should indicate state 1 for the top figures and state 2 for the bottom ones.

9. In table 1, you should point out and explain why the 4-4 matrices are symmetric.

10. In tables 3 and 4, please provide the explicit formulas for the computation of those matrices.

---

## Author Comment (AC1) · 5 Jun 2018

**Reviewer #1's comments and our responses**

1. The reviewer commented that the method description is not enough for the general audience of *Climate of the Past*. The reviewer pointed out that the model comparison (Sec 3.3) with Figure 5 is confusing. The Reviewer #2 also suggested adding more description about the HMM (comment 5) and comparison to the classical correlation analysis (comment 1). We can add more general explanations about the method and also modify Section 3.3 as follow:

Statistical modeling involves developing relationships between one set of predictor variables and another set of predictands. Paleoclimate reconstructions likewise develop models to relate proxy information (predictors) to past climate variables (predictands). Thus, statistical modeling and paleoclimate reconstructions both seek the same goals, and approaches of varying complexity are found to infill missing data or to understand relationships among variables.

In paleoclimate studies, as in any set of observations, not all important variables can be observed or reconstructed. It is typical in such situations to hypothesize linkages among observed variables, but a more direct observation of the mechanism involved in the linkage are not recorded. So, one might expect that A causes B that causes C, but only A and C are observed. Statistical modeling can help identify or quantitatively assess relationships between A and C, even in the presence of hidden variables such as B.

The autoregressive (AR) approach adds value by allowing the state at previous times to be among the predictors of the present state predictands. Typically, causes precede effects, so the AR approach allows for an interpretation of causality--if a predictor precedes the predictand in time, then it is the cause rather than *vice versa*. Simultaneous correlations among variables are frequently interpreted as implying causality, but they can represent a number of relationships--cause and effect, effect and cause, or accidental correlations without causal relationships. The greater precision of the AR models allows for examination of causal relationships under the assumption of cause preceding effects.

Furthermore, the HMM provides a quantitative justification of transitions between different epochs governed by regime shifts in the surrounding climate. Even though these shifts might not be directly detectable in any of the recorded variables alone, the HMM provides a technique that allows all variables to contribute equally in identifying shifts in the relationships among the variables.

Section 3.3:
The HMM is a special case of the AR-HMM; The AR-HMM with zero autocovariance term ($\theta_{s(t)}$) is identical to the HMM. So, if the AR-HMM results in the proxies having weak autocorrelation, $\theta_{s(t)}$ should be close to zero, and the other parameters of the AR-HMM (the noise covariance matrices ($\Sigma_{s(t)}$)) will resemble their equivalents in the HMM. Thus, were the HMM an adequate model to describe the proxy data, then allowing the extra degrees of freedom

in the AR-HMM would result in little extra predictive power, and this result would not change the interpretation of the data from the interpretation found using the HMM alone. However, in this particular dataset, the AR-HMM resulted in extremely large auto-correlation relationships (the entries of the estimated $\theta_{s(t)}$ are close to one) and furthermore the other model parameters (the estimated noise covariance matrices) are quite different between the HMM and the AR-HMM. Fig. 5 visualizes and compares the estimated $\theta_{s(t)}$ of the HMM and AR-HMM. The fact that the AR-HMM coefficients do not resemble the HMM in pattern, magnitude, or implied relationships means that a dependence of the data on values at a previous time is a critical aspect of the data. Thus, a key conclusion from the statistical models is that the past values of each proxy predicts its own proxy variability better than the different proxy-to-proxy cross-correlations at the same time (or indeed the cross-correlations among past and present values). This fact implies that the different proxies in this particular dataset are not causally related to one another, as is often assumed in multi-proxy paleoclimate analyses (e.g., Hu et al. 2017). This result probably does not apply to all muli-proxy records, indeed many are probably causally linked, but our methodology for testing that assumption by comparing HMM to AR-HMM is generic. Thus, in this location, the four proxies (*SST; $C_{37}$; $\partial^{15}N$; %N*) are not related to each other in the local sense that variability in any one dominates or contributes significantly to variability in another through a local physical or biological mechanism.

2. The reviewer commented that a) the Uk'37 proxy may be biased in the region toward warm temperatures and that b) associations between alkenone-based temperatures and productivity might be erroneously interpreted.

We appreciate these concerns and responded by moving proxy information into the Method section and addressing the referees concerns there. In brief, we can cite two substantial data sets that look at the alkenone proxy in the Eastern Equatorial Pacific and argue that there is no indication of a SYSTEMATIC bias relative to mean annual SST in the region

We can evaluate the second claim, as we report Uk'37 unsaturation, bulk organic nitrogen and C37total, an index of the sediment concentration of alkenones. As we demonstrate, the index of bulk phytoplankton production and C37 total are significantly correlated, suggesting that haptophyte production indeed follows total ecosystem production. And furthermore, the lack of a strong coupling between the Uk'37 index and either productivity proxy--as found by the statistical methods used in this paper--argues against the existence of the production-SST bias suggested by the reviewer. The fact that an inorganic proxy (opal, as reported by Chazen et al.) does not resemble the organic proxies can most likely be explained by variations in the preservation of opal, a notorious confounding influence on interpreting that proxy quantitatively

We now include additional text in 2.1 (Data Collection):

The four records examined are proxies for sea surface temperature (*SST*) through the alkenone proxy, biological productivity of a specific phytoplankton group (*$C_{37}$*) through analyses of the

abundance of alkenones (representing haptophyte algal productivity), subsurface properties through analyses of $\partial^{15}N$, an index of subsurface oxygenation and denitrification, and the percentage of organic nitrogen (%N) which is a composite of all biological inputs to the sediment. We interpret the alkenone Uk'37 index as an approximation to mean annual sea surface temperature. Although anomalies Uk'37 values have been reported in the region (Prahl et al., 2010; Kienast, 2012), there is no convincing evidence for seasonal bias based on analyses of modern sediments over a broad region of the Eastern Equatorial Pacific with very strong gradients in the timing of maximum annual biological production (Kienast et al., 2012; Timmerman et al., 2014). Analyses of modern sediments in the region conducted at the Brown University laboratory show agreement with mean annual temperatures in the region of our core study to within the standard empirical proxy calibration (e.g. subset of data reported in Kienast et al, 2012). Our paleo-productivity interpretations are guided by the presence of a proxy that responds to total phytoplanktion production (%N) and to a subset of the haptophyte production (C37total); we can therefore assess whether alkenone production is coupled or decoupled to a generalized biological response over time.

3. The reviewer suggested considering other factors, such as laminations bioturbation, and sediment mixing, to explain more about the decadal predictability.

The reviewer makes a good point about the potential down-core differences in variance being driven by variations in oxygenation and bioturbation. In some sense this is a chicken and egg question, because the existence of laminations is in fact coupled to some of the variables represented by our proxies, such as density stratification and organic matter flux. It is therefore difficult to assess whether the presence/absence of laminations is a confounding factor or part of the oceanographic signal represented in our time series.

However, when we compared a visual index of lamination/bioturbation, based on X-radiographs of the core, we see the following results. In the HMM, the "calm" state is associated with a significantly more negative d15N value, consistent with, although not proof of, a preferential smoothing of variance in non-laminated intervals. However, this association does not persist in the AR-HMM results, suggesting that this 2-state model does not reflect a preservational bias of variance.

We thank the reviewer for sharpening our analysis in this regard, and have modified the text in Section 3.3 by adding:

A caveat arises in assessing variance in the time series: changes in the extent of laminations down-core, which could introduce differential smoothing of the results. We can assess the possible influence of lamination versus bioturbation in two ways: a visual comparison of X-radiographs of the core, which show the presence/absence of laminations, and comparison to d15N, which is strongly indicative of lamination (high d15N signifies intense depletion of oxygen in the subsurface). The results of the HMM and AR-HMM differ significantly in this regard. The presence of State 1 versus State 2 correlates strongly with the degree of lamination/d15N

proxy in the case the HMM model (the "noisy" state occurring much more frequently in laminated intervals). This association is confirmed by the significant offset in the mean values of d15N for State 1 and 2 (Table 1). However, the AR-HMM removes any significant dependence on the occurrence of the "noisy" versus "calm" states on the status of lamination down-core, and is confirmed by the negligible offset in the mean d15N reported for the two states (Table 2).

---

## Author Comment (AC2) · 5 Jun 2018

**Reviewer #2's comments and our responses**

The reviewer suggested improving the model descriptions:

1. The reviewer suggested adding a paragraph to describe the disadvantages of the classical correlation analysis. See the reply to the Reviewer #1's first comment. We wrote a paragraph that describes the advantages of the method we used in the study and the disadvantages of the correlation analysis (see above).

2. Reviewer's comment: *In the introduction, you should explicitly state that the four proxies come from the same core, as it is an important implicit hypothesis.*
   In Section 2.1 Data Collection, we state that "... a sediment core retrieved from the central Peru margin". We can also modify the second sentence of Section 1.0 Introduction: "The sediment record is retrieved from a single sediment core located at the central Peru margin and sampled as high-resolution to …"

3. The reviewer suggested describing the age model of the core.
   The age model from [1] is used to estimate the age of the core top and base. We did not estimate the age for each sample because we assume a constant accumulation rate. The assumption does not affect the HMM and AR-HMM analysis significantly, because the sediment record is retrieved from a single sediment core so cross-correlations are not affected significantly by age. Modest variations in accumulation would only weakly affect the accuracy of the autocorrelations, as those are linkages assessed only between neighboring values of the data rather than long sections of the core. If the constant accumulation rate assumption is not valid, then we would consider either direct age assignments via C-14 dating ([2]) or indirect age assignments via synchronization based on a global climate variable, such as glacial ice volume using the benthic $\square^{18}O$ proxy ([3], [4]). However, the age modeling efforts thus far do suggest that the site has a high and steady sedimentation rate across the Holocene. Therefore, we assumed a constant rate and used age estimates only for the top and bottom of the core.

4. The reviewer suggested describing the results when the number of states is assumed to be one or more than two. The goal of this study is to reveal the hidden regime changes under the sampled data, so we assumed that the number of states is two or more than two. We experimented during the early stages of the project by assuming that there might exist four states, which include two additional states as intermediate states, and found that the regime changes through the intermediate or transitional states was very fast, as shown the figure below. As this methodology does not exhibit persistence in the intermediate states, it is not meaningful to consider those intermediate states as separate regimes. This approach would also have indicated if there were three or four sustained regimes (i.e., nothing about the approach implied that these extra third and fourth states had to be transitional, but that was what was found). Therefore, we proceeded on to all later calculations assuming the number of states to be two, hence the two-state HMM and AR-HMM.

[Figure]

This figure shows the probability of being each state when we assumed four states, including 2 intermediate states. The state can change from 1 to 4 through 2u and 3u or directly from 1 to 4. Also, the state can change from 4 to 1 through 3d and 2d or directly from 4 to 1. Most state changes occur without intermediate states.

5. The reviewer suggested adding more description of the HMM. See the reply to the Reviewer #1's first comment as well. The following paragraph can be added as an introduction of HMM and AR-HMM as well.

Both HMM and AR-HMM consist of observed data *X(t)* and two kinds of hidden states *s(t)*. The measured data from the sediment core correspond to *X(t)*, and the unobserved state for each observed data corresponds to *s(t)*. The unobserved hidden states are analogous to the terms "regimes" that are described in climate studies. The states are hidden because they are to be determined from relationships within the data by the model, rather than indicated directly, e.g., if the value of one variable indicated which regime the data was in at any given time.

The figure below illustrates the dependencies among hidden states (S) and observed data (X) of the two models. State dependencies are the same in both models. Both models

have hidden states that have the Markov property, meaning that the future state does not depend on the past states given the present states. The difference between the two models is the dependency between observations (X) that are adjacent in time to each other (X(1) to X(2)). In the HMM, a current observation is solely dependent on present observations and the current state. The HMM assumes that a current observation follows the normal distribution with means and variances determined by its state. Thus, a current set of observations is independent of other sets of observations at other times, although its state does depend on what state was determined at a previous time. In the AR-HMM, a current observation depends not only on a current and previous state but also on the previous observations. Therefore, the AR-HMM model allows for examination of causal relationships among the observed variables--inferring connections beyond just "regime" shifts and into relationships such as SST predicts productivity at at later time.

[Figure]

(a) HMM                              (b) AR-HMM

Dependencies among observations X(t) and hidden states S(t) for (a) HMM and (b) AR-HMM. Nodes are connected with an arrow if one node at the head of an arrow depends on another node at the origin of an arrow.

Also, after the equation (1), we can add the following statement to explicitly state that the equation (1) is for both states: "This equation represents the two-state AR-HMM, where the state value $s(t)$ can be either 1 (Noisy state) or 2 (Calm state)."
The reviewer also suggested investigating other methods that do not require the interpolation procedure. There exist some methods that can be applied to irregularly sampled data without the interpolation procedure, as the reviewer commented. However, those methods are not easily applicable when we consider multiple regimes and regime changes. In this study, we used the first order autoregressive model, meaning the model includes one immediately preceding value. To use this model, we need an estimated values for each missing observation value. Therefore, we estimated the expectation of the missing values by using the Kalman filter. To develop a method to analyze irregularly spaced samples with regime changes but without interpolations would be an interesting topic for future study, but is outside the scope of this paper and probably also the audience of this journal.

6. The reviewer suggested adding more statistical analysis for the conclusion that the four proxies are not causally related. If the four proxies are causally related, then the off-diagonal entry values

of the autocovariance matrix should be similar to the diagonal entry values. If the off-diagonal entry values and diagonal entry values are comparable, then further analysis would be required. However, the difference in absolute values between diagonal and off-diagonal values are extremely large; See the estimated autocovariance regression matrix ($\theta$) values in Table 2. The smallest value of the diagonal entries is 13 times larger than the largest value of the off-diagonal entries, so we did not conduct a further statistical analysis. Also, it is not desirable to apply a frequentists approach when using a Bayesian model. Ideally, we would prefer a full Bayesian model comparison. However, that requires summing and integrating over all the unknowns which is not possible in this because there exist too many interconnections in the model parameters to complete all the sums and integrals simultaneously.

Or, to put it more simply, the AR-HMM results are not quantitatively different from the HMM, they are qualitatively different. The dominant relationships change entirely, not just slightly or within the range of statistical uncertainty, the whole nature of the AR-HMM and HMM models is different. We note again that this is not expected, unless the additional degrees of freedom of the AR-HMM system have systematically allowed for different behavior. This change is the nature of "Granger causality", i.e., if the system changes qualitatively when past values are allowed as predictors, it implies that same-time correlations cannot be causations.

7. Reviewer's comment: *In Fig. 2, 3 and 4, please draw in function of the time/age rather than the time step.*
The sediment record is retrieved from a single sediment core located at the central Peru margin. Also, the site of the sample has a high and steady sedimentation rate across the Holocene, and the samples are obtained from 2cm slices taken every 5cm. Therefore, we assumed a constant accumulation rate. Because we assumed a constant accumulation rate, we did not use additional age model to estimate the age for each sample and used the time step instead of the time or age. The accumulation rate does not affect the cross-correlations or autocorrelations directly.

8. Reviewer's comment: *In Fig. 5, you should indicate state 1 for the top figures and state 2 for the bottom ones.*
The top and bottom rows represent the correlation matrices of the state 1 and state 2, respectively. We can add this information to the figure and also the figure description.

9. Reviewer's comment: *In table 1, you should point out and explain why the 4-4 matrices are symmetric.*
The 4-by-4 matrix in Table 1 is the covariance matrix of the error term. The $(i,j)$ entry of the covariance matrix is the covariance of the $i$th entry and $j$th entry. Because the covariance is commutative, meaning that $cov(\varepsilon_i, \varepsilon_j) = cov(\varepsilon_j, \varepsilon_i)$, the covariance matrix is symmetric. It can be seen from the definition of the covariance matrix as well. The $(i,j)$ entry of the covariance matrix is defined as $\Sigma_{ij} = cov(\varepsilon_i, \varepsilon_j) = E[(\varepsilon_i - E[\varepsilon_i])(\varepsilon_j - E[\varepsilon_j])]$. Therefore, $\Sigma_{ij}$ is equal to $\Sigma_{ji}$, and the matrix is symmetric ([5]).

10. Reviewer's comment: *In tables 3 and 4, please provide the explicit formulas for the computation of those matrices.*

The correlation matrix is defined as $corr(x_i, x_j) = \frac{cov(x_i, x_j)}{\sigma_i \sigma_j}$ , where *cov* means covariance, and $\sigma_i$ and $\sigma_j$ are the standard deviations of $x_i$ and $x_j$ , respectively.

References
[1] Chazen, C. R., M. A. Altabet, and T. D. Herbert (2009), Abrupt mid‑Holocene onset of centennial‑scale climate variability on the Peru‑Chile Margin, *Geophys. Res. Lett.*, 36, L18704, doi:10.1029/2009GL039749.
[2] Blaauw, M. and Christen, J.A., 2011. Flexible paleoclimate age-depth models using an autoregressive gamma process. *Bayesian analysis*, *6*(3), pp.457-474.
[3] Lin, L., Khider, D., Lisiecki, L.E. and Lawrence, C.E., 2014. Probabilistic sequence alignment of stratigraphic records. *Paleoceanography*, *29*(10), pp.976-989.
[4] Ahn, S., Khider, D., Lisiecki, L.E. and Lawrence, C.E., 2017. A probabilistic Pliocene–Pleistocene stack of benthic δ18O using a profile hidden Markov model. *Dynamics and Statistics of the Climate System*, *2*(1).
[5] Wackerly, D., Mendenhall, W. and Scheaffer, R., 2007. *Mathematical statistics with applications*. Nelson Education.

---

## Author Comment (AC3) · 5 Jun 2018

Thank you for taking time to carefully read and evaluate our manuscript. We appreciate the comments and suggestions made by reviewers.

Reviewer #2 suggested considering another journal, but we still think this journal, *Climate of the Past*, is the right place to discuss statistical analysis results of paleoclimate data. There is no journal specializing in statistical paleoclimate analysis, and this study applies standard statistical methods to paleoclimate data, so a purely statistical journal is not appropriate. It is our hope that work such as this one can find a home in *Climate of the Past*, as we support the open-access and non-profit nature of that journal and believe statistical work is an important development in paleoclimatology.

Accordingly we have significantly modified our introduction to emphasize the utility of our approach for paleoclimatic analyses and revised the text to give the non-statistical reader a more intuitive grasp of our interpretations.

Modified Introduction:

Paleoclimatic time series hold the promise to extend our knowledge of oceanic, atmospheric, and ecological variability on the timescale of decades to centuries, a time window poorly constrained by instrumental observations.  A frequent assumption of such studies is that significant modes of variability detected in the historical record (ENSO- El Nino/Southern Oscillation, PDO- Pacific Decadal Oscillation, AMO- Atlantic Meridional Oscillation, etc.) persist into the past and there may also exist other fluctuations detectable only through the paleoclimate record, but that resemble modern patterns (Knudsen et al., 2011; Koutavas and Joanides, 2012; Newman et al., 2016).  Such patterns often involve an ensemble of couplings between aspects such as pressure gradients, surface temperature, and biological productivity--all of which might be observed by sediment proxy methods--and couplings or correlations might be used to infer that the underlying variability does in fact resemble documented modes of modern internal climate variability.

This paper examines statistical aspects of a long-duration, high-resolution, multi-dimensional time series that record variations in among sea surface temperature ( SST), phytoplankton productivity, and intensity of the oxygen minimum zone (OMZ) over the Holocene epoch (0.60 to 9.44 kA B.P.) along the central Peru margin, a region strongly affected by the modern ENSO and PDO cycles (Brink et al., 1983; Newman et al., 2016). The sediment is sampled at high-resolution to amount to roughly 3-year averages sampled every 7 years under the accumulation rate typical of the region, and the different proxies are analyzed using the same core sampling, so that correlations are robust of age model uncertainties. These records indicate both surface and subsurface variability in the physical and biological state. Simple correlation analyses revealed that proxy associations vary over time; in some intervals following expected ENSO-like correlations of low sea surface temperature (SST) and enhanced biological productivity; in other intervals this correlation is not apparent (Chazen et al., 2009).  We develop here a statistical framework to analyze the evolving relationships over the Holocene in order to  interpret proxy correlations, characteristic  timescales of variability ("regimes"), and predictability.

Some of the key questions that may be addressed by time series analysis in this region are whether the variability arises from a local or internal source, such as variation in physics through mixing or eddies at the surface (Brink et al., 1983, Colas et al. 2012) or changes in the biological makeup of ecosystems in the region (e.g., Gooday et al. 2010), or from a remote or external source, such as variations in the water properties arriving at the site through large scale modes such as El Nino or the Pacific Decadal Oscillation (Mantua et al. 1997, Deser et al. 2010). The site (Fig. 1) is known for wind-driven upwelling (Brink et al. 1983) at depths shallower than 250m and low oxygen concentrations at depth typical of the eastern tropical South Atlantic oxygen minimum zone, which has been highly variable near 250m depth in recent times (Stramma et al. 2008). Despite the low oxygen levels at depth, the typical sediment accumulation rate over the Holocene during these samples is high (70 cm/kyr), which suggests high, sustained biological productivity and presumably a persistent level of oxygen demand.

A visual analysis of the proxy records (Fig. 2) suggests that the variability of four proxies might fall into multiple regimes: one state with high variability and another state with low variability. This *biphasic* behavior guided our initial analysis using a Hidden Markov Model (HMM; Rabiner, 1989). Hidden Markov methods are increasingly used as a statistically robust automated method for identifying climate regime shifts (e.g., Majda et al. 2006, Franzke & Woollings 2011, Ahn et al. 2017). A benefit of our approach is that it can objectively identify regimes of paleoclimatic behavior in which correlations between proxies and proxy variance evolve (and perhaps alternate) over time. We also explored the possibility of more than two states, but found that these extra regimes were visited only transiently, so parsimony suggested retaining only two modes.

A less common tool in climate modeling is the autoregressive hidden Markov method (AR-HMM, Hamilton, 1988, 1989, 1994) which allows for some memory in the system through a dependence on previous proxy values as well as correlations in the present proxy value noise. The application of this method here, and the insights gained from this application, are a key breakthrough found in our analysis. Both our HMM and AR-HMM results show that there exist two regimes of variability in proxy space at site MW8708-PC2. Here the AR-HMM technique is used to probe deeper into distinctions between causality and correlation, under the premise that a predictive cause should precede its effect in time. As the HMM method examines only simultaneous-in-time correlations, it is not capable of distinguishing causation from correlation in this way. A surprising result of this study is that our conception of the relationships among these proxies from the HMM analysis changed dramatically when the AR-HMM technique was applied and contrasted to the more standard HMM approach. The AR-HMM shows that both climatic regimes show high auto-correlation and low cross-correlation, thereby indicating that none of the proxies are good predictors of other proxies on interannual to decadal timescales. Thus, a hypothesis of local causality between the variables, such as mixing driving local productivity, is not supported by the AR-HMM analysis. This lack of causality is robust to the biphasic regime shifts as well, although in a different location where regime change is not present, a simpler autoregressive only approach can be used to assess causation versus correlation following a similar approach to the methods used here. The software provided with this paper (https://github.com/seonminahn/ARHMM) can be applied for the analysis of multi proxy data from a core record.